# Rapid, automated, and experimenter-free touchscreen testing reveals reciprocal interactions between cognitive flexibility and activity-based anorexia in female rats

Kaixin Huang[1,2†], Laura K Milton[1,2†], Harry Dempsey[1], Stephen J Power[1], Kyna-Anne Conn[1,2], Zane B Andrews[1,2], Claire J Foldi[1,2]*

[1]Department of Physiology, Monash University, Clayton, Australia; [2]Monash Biomedicine Discovery Institute, Clayton, Australia

*For correspondence:
claire.foldi@monash.edu

†These authors contributed equally to this work

Competing interest: The authors declare that no competing interests exist.

**Abstract** Anorexia nervosa has among the highest mortality rates of any psychiatric disorder and is characterized by cognitive inflexibility that persists after weight recovery and contributes to the chronic nature of the condition. What remains unknown is whether cognitive inflexibility predisposes individuals to anorexia nervosa, a question that is difficult to address in human studies. Our previous work using the most well-established animal model of anorexia nervosa, known as activity-based anorexia (ABA) identified a neurobiological link between cognitive inflexibility and susceptibility to pathological weight loss in female rats. However, testing flexible learning prior to exposure to ABA in the same animals has been thus far impossible due to the length of training required and the necessity of daily handling, which can itself influence the development of ABA. Here, we describe experiments that validate and optimize the first fully-automated and experimenter-free touchscreen cognitive testing system for rats and use this novel system to examine the reciprocal links between reversal learning (an assay of cognitive flexibility) and weight loss in the ABA model. First, we show substantially reduced testing time and increased throughput compared to conventional touchscreen testing methods because animals engage in test sessions at their own direction and can complete multiple sessions per day without experimenter involvement. We also show that, contrary to expectations, cognitive inflexibility measured by this reversal learning task does not predispose rats to pathological weight loss in ABA. Instead, rats that were predisposed to weight loss in ABA were more quickly able to learn this reversal task prior to ABA exposure. Intriguingly, we show reciprocal links between ABA exposure and cognitive flexibility, with ABA-exposed (but weight-recovered) rats performing much worse than ABA naïve rats on the reversal learning task, an impairment that did not occur to the same extent in rats exposed to food restriction conditions alone. On the other hand, animals that had been trained on reversal learning were better able to resist weight loss upon subsequent exposure to the ABA model. We also uncovered some stable behavioral differences between ABA susceptible versus resistant rats during touchscreen test sessions using machine learning tools that highlight possible predictors of anorectic phenotypes. These findings shed new light on the relationship between cognitive inflexibility and pathological weight loss and provide targets for future studies using the ABA model to investigate potential novel pharmacotherapies for anorexia nervosa.

## Editor's evaluation

This important manuscript describes a fully automated touchscreen cognitive testing system for rats that reduces the length of training required to learn a task and eliminates the need for daily handling. These features make it possible to assess cognitive behaviors in conjunction with other

neurobehavioral paradigms during adolescence, an important advance in the field. The data are compelling in showing that cognitive flexibility does not promote susceptibility to severe weight loss in the activity-based anorexia (ABA) paradigm and support the claim that the cognitive deficits seen in ABA-exposed rats reflect an important clinical feature of the pathophysiology of anorexia nervosa.

## Introduction

Cognitive flexibility refers to the capacity to modify behavioral choices to meet the demands of a changing environment and is crucial for selecting appropriate responses based on context and circumstance (*Diamond, 2013*). Impairments in cognitive flexibility are common to a range of psychiatric illnesses including schizophrenia (*Floresco et al., 2009*; *Thai et al., 2019*), obsessive-compulsive disorder (*Gruner and Pittenger, 2017*), and substance use disorders (*Smith and Laiks, 2018*; *Sampedro-Piquero et al., 2019*), which are characterized by stereotypical patterns of rigid behaviors that persist despite negative consequences, ultimately impacting decision-making. Individuals with a current or previous diagnosis of anorexia nervosa also exhibit rigid behaviors, especially surrounding illness-relevant stimuli such as feeding and exercise (*Tchanturia et al., 2012*; *Tchanturia et al., 2004*; *Galimberti et al., 2013*; *Roberts et al., 2010*; *Steinglass et al., 2006*). While impaired cognitive flexibility is most severe in patients acutely ill with anorexia nervosa and likely contributes to perpetuating the condition (*Tchanturia et al., 2012*; *Roberts et al., 2010*), the persistence of inflexible behavior following weight recovery and in unaffected sisters of patients with anorexia nervosa suggests that it is involved in the etiology of the disorder (*Roberts et al., 2010*; *Steinglass et al., 2006*; *Tenconi et al., 2010*). What remains to be determined is whether cognitive inflexibility itself predisposes individuals to develop anorexia nervosa and could be used as a biomarker to predict illness onset or severity in individuals at risk. Moreover, a detailed understanding of the neurobiology underlying an inflexibility that persists after weight recovery in individuals with anorexia nervosa is imperative to develop novel pharmacotherapies that can aid in long-term recovery (*Tchanturia et al., 2011*; *Duriez et al., 2021*; *Errichiello et al., 2016*).

While the premise that cognitive rigidity is a fundamental trait of anorexia nervosa is well-accepted, studies assessing cognitive flexibility in patient populations are prone to inconsistent findings, which are likely amplified by large discrepancies in participant demographics and experimental approaches (*Miles et al., 2020*). It is also difficult to determine from human studies the neurobiological mechanisms that precede the development of anorexia nervosa that could act as targets for early intervention. The question then arises - how can we assess the neural mechanisms of cognitive flexibility in animal models that adequately capture the clinical presentation in anorexia nervosa patients? Rodent models that incorporate key aetiological features, such as the most well-established animal model of anorexia nervosa known as ABA, have been instrumental in identifying the specific neural circuits that contribute to disordered cognitive functioning (*Milton et al., 2021*). Additionally, the last decade has witnessed an explosion in the availability of innovative tools including optogenetics (*Tye and Deisseroth, 2012*), chemogenetics (*Roth, 2016*), and calcium imaging (*Lütcke et al., 2010*), to manipulate and record neural activity in freely behaving animals. These approaches give an unprecedented ability to answer questions about the relationship between brain function and behavior relevant to a range of human disorders, including anorexia nervosa. However, the interest in new techniques to modify and record brain function has not been matched with adequate enthusiasm regarding the quantification and analysis of behavioral outputs that are critical for the assessment of these relationships.

With this in mind, the study of cognition and behavior in rodents has benefited in recent years from advances in technology that have increased the translational capacity of rodent models of human pathologies (*Keeler and Robbins, 2011*; *Bussey et al., 2008*). A major contribution to improving translation has been the incorporation of touchscreens displaying visual stimuli in rodent test batteries that closely mimic those used for human cognitive testing (*Bussey et al., 2008*), which improves the standardization and interpretation of data. However, touchscreen testing in rodents has thus far required significant time and experimenter intervention to transfer subjects to and from the testing chamber. Indeed, it is well known that experimenter involvement influences experimental outcomes, particularly so for behavioral studies - including those involving the ABA model, in which the outcomes are known to be influenced by experimenter handling (*Carrera et al., 2006*). Along with stress from handling, which varies between experimenters and, therefore, differentially impacts upon

task performance (*Deutsch-Feldman et al., 2015*; *Sorge et al., 2014*; *Meijer et al., 2007*), manual transfer to test chambers at times that suit the experimenter is insensitive to the current motivational state of the animal. Thus, while the wide adoption of touchscreen cognitive testing has already yielded substantial benefits for behavioral neuroscience, the next frontier lies in the automation of the role of the experimenter in gatekeeping touchscreen access (*Rivalan et al., 2017*; *Winter and Schaefers, 2011*).

One approach has been to relocate the operant testing modules to inside the home cage for quantification of complex operant and feeding behaviors (*Matikainen-Ankney et al., 2021*), or to connect individual operant test chambers to the home cage by way of a short tunnel (*Bruinsma et al., 2019*) to minimize intervention and provide a higher throughput training-testing framework. However, these both have a requirement for animals to remain socially isolated to ensure that the cognitive performance of each individual can be monitored over time. Considering that social isolation itself can induce cognitive deficits (*Bianchi et al., 2006*; *Hemmer et al., 2019*) and depression-like behavior (*Ieraci et al., 2016*; *Fone and Porkess, 2008*), this is a huge confound for the assessment of cognition in rodent models. In contrast, appropriate social interaction can enhance neuroplasticity (*Liang et al., 2019*), emotional and social intelligence (*Torquet et al., 2018*) and influence performance on complex cognitive tasks (*Nagy et al., 2020*). Recently, the capacity to monitor and track rodents in social groups has become achievable with radiofrequency identification (RFID) technology (*Peleh et al., 2019*; *Redfern et al., 2017*) in combination with gating access to test chambers based on a method of automatic animal sorting (*Winter and Schaefers, 2011*; *Caglayan et al., 2021*; *Kaupert et al., 2017*). The development of a fully automated, experimenter-free method for touchscreen-based cognitive testing in rats has been ongoing since the first prototype was constructed in 2017, allowing the successful adaptation of the trial-unique non-matching to location (TUNL) task in an environment that both eliminates experimenter intervention and allows animals to live in social groups throughout testing (*Rivalan et al., 2017*). This study demonstrated that the learning rate of self-motivated and undisturbed rats was much faster when experimenter involvement is removed.

The potential to more rapidly test cognition in rodents without experimenter intervention and in social groups opens the door to examine whether cognitive inflexibility predisposes individuals to pathological weight loss in ABA – particularly important because the ABA model develops differently in adult compared to adolescent ages (*Beeler and Burghardt, 2021*). It also allows us to determine the persistence of inflexibility following weight recovery in ABA rats, in order to use this model to screen novel pharmacotherapeutics for anorexia nervosa. In the present study, we used the automated and experimenter-free touchscreen testing system developed from the prototype mentioned above (and now commercially available from PhenoSys, GmbH) to investigate both of these ideas. This automated approach also enables animals to express a more naturalistic behavioral repertoire, a feature ideally suited to comprehensive interrogation with unbiased machine learning approaches to quantify behavioral profiles. Here, we exploited this union with analysis of uninterrupted video recordings of touchscreen sessions using DeepLabCut (*Mathis et al., 2018*) and B-SOiD (*Hsu and Yttri, 2021*) to determine the behavioral drivers of cognitive performance. Moreover, the high-throughput pipeline for video analysis from touchscreen sessions that we have established is available openly and may prove useful for future experiments aimed at identifying the behavioral correlates of cognitive performance in rodent models.

## Materials and methods

### Key resources table

| Reagent type (species) or resource | Designation | Source or reference | Identifiers | Additional information |
|---|---|---|---|---|
| Strain, strain background (*Rattus norvegicus*, female) | Sprague-Dawley | Monash Animal Research Platform | n/a | |
| Software, algorithm | GraphPad Prism 9.1.1 | GraphPad Software | RRID:SCR_002798 | |
| Software, algorithm | Scurry Activity Monitoring Software | Lafayette Instruments, IN | Model 80859 S | https://lafayetteneuroscience.com/products/scurry-activity-software/ |

*Continued on next page*

*Continued*

| Reagent type (species) or resource | Designation | Source or reference | Identifiers | Additional information |
|---|---|---|---|---|
| | | *Mathis et al., 2018* | | |
| | | *Nath et al., 2019* | | |
| Software, algorithm | DeepLabCut | | Version 2.2.1.1 | https://github.com/DeepLabCut/DeepLabCut |
| Software, algorithm | B-SOiD | *Hsu and Yttri, 2021* | Version 2.0 | https://github.com/YttriLab/B-SOID |
| Software, algorithm | MultiCam Capture | Pinnacle Studio | | https://www.pinnaclesys.com/en/products/multicam-capture/ |
| Software, algorithm | PhenoSoft | PhenoSys, GmbH | | |
| Other | 2.5 mm diameter sucrose pills | Homeopathic Supply Company | Batch number: 08753017 | https://www.hsconline.co.uk/collections/tablets/products/2-5mm-diameter-sucrose-pillules |
| Other | Lab Bedding Products | The Andersons | Pure-o-cell | https://andersonsplantnutrient.com/engineered-products/lab-and-enrichment/products/pure-ocel |
| Other | Laboratory Enrichment | The Andersons | Enrich-n'Nest | https://andersonsplantnutrient.com/engineered-products/lab-and-enrichment/products/enrich-nnest |

## Animals and housing

All animals were obtained from the Monash Animal Research Platform, Clayton, VIC, Australia. Initial exploration and optimization of the novel touchscreen testing system were performed in a cohort of female Sprague-Dawley rats (n=20), 6–7 weeks old at the commencement of testing. To assess both ABA and cognitive behavior in the same animals, female Sprague-Dawley rats were 5–6 weeks of age upon arrival in the laboratory. Young female rats were used in these studies because they are particularly vulnerable to developing the ABA phenotype, a feature that is incompletely understood but has translational relevance to the increased prevalence of anorexia nervosa in young women. Animals were group-housed and acclimated to the 12 hr light/dark cycle (lights off at 1100 hr) for 7 days before experiments commenced. To examine whether cognitive flexibility predicted pathological weight loss in ABA, rats (n=30) were tested on the pairwise discrimination and reversal learning task and subsequently exposed to the ABA paradigm. To determine whether exposure to ABA altered cognitive performance on the same task, rats (n=24) were exposed to the ABA paradigm and allowed to recover to ≥100% body weight prior to pairwise discrimination and reversal learning (see *Figure 1* for the timeline of experiments). To control for the effects of prior food restriction on cognitive ability in these tasks, a separate cohort of animals (n=22) were exposed to a period of restricted food access that matched the ABA exposure timeline. Rats were age-matched for the initiation of ABA exposure, to ensure there were no effects of age on vulnerability to weight loss (see *Figure 3—figure supplement 2*).

Because both wheel running and food intake are known to fluctuate with the estrous cycle in female rats (*Anantharaman-Barr and Decombaz, 1989*), a male rat was housed in all experimental rooms at least 7 days prior to experimentation in order to facilitate synchronization of cycling, known as the Whitten Effect (*Cora et al., 2015*). All experimental procedures were conducted in accordance with the Australian Code for the care and use of animals for scientific purposes and approved by the Monash Animal Resource Platform Ethics Committee (ERM 29143 and 15171).

## Automated sorting and touchscreen testing using PhenoSys multimodal apparatus

Rats were anaesthetized with isoflurane in oxygen (5% for initiation, 2.5% for maintenance) and subcutaneously implanted with RFID transponders (2.1 × 12 mm; PhenoSys, Berlin) into the left flank using a custom-designed syringe applicator. The incision site was sealed by tissue adhesive (Vetbond 3 M; NSW, Australia). Following RFID implantation, rats were group housed (n=6 per group) in separate home cages of the automated touchscreen testing apparatus (*Figure 1—figure supplement 1*; PhenoSys, Berlin) and allowed to habituate to the home cage with ad libitum food access for 1 day

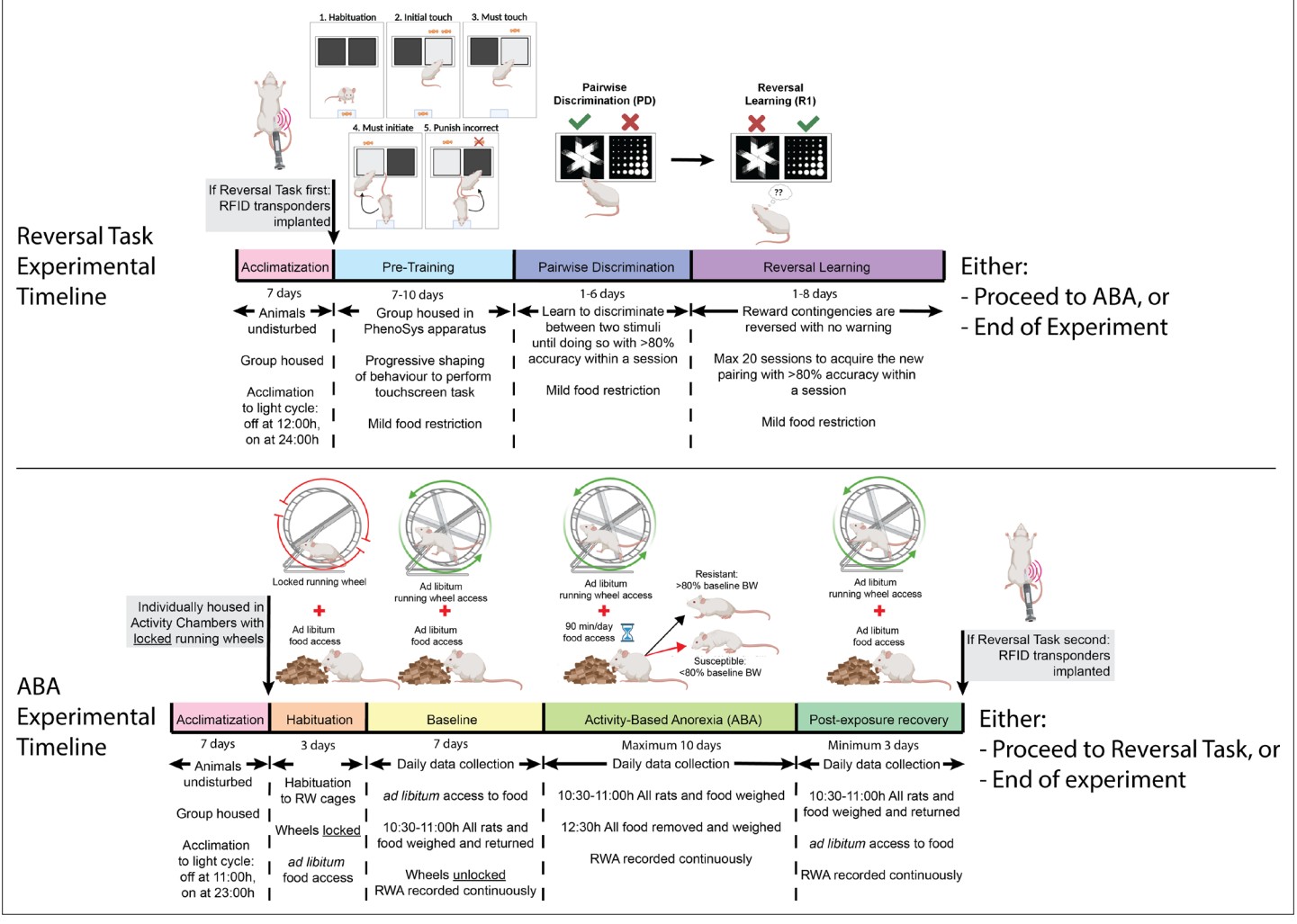

**Figure 1.** Timeline of experiments. Rats were underlined{acclimated} to the reversed light cycle for 7 days before each experiment commenced. In experiment 1, rats underwent touchscreen cognitive testing before undergoing the activity-based anorexia (ABA) paradigm. In experiment 2, rats were exposed to the ABA paradigm prior to cognitive testing in the PhenoSys apparatus. The PhenoSys cognitive testing paradigm consisted of 7–10 days of pre-training, 1–6 days of pairwise discrimination, and 1–8 days of reversal learning. The ABA paradigm consisted of 3 days of habituation with ad libitum food access, 7 days of baseline testing with ad libitum access to food and a maximum of 10 days of ABA conditions with time-limited food access (90 min/day) and unrestricted running wheel access, followed by a minimum of 3 days of body weight recovery with reinstatement of ad libitum access to food.

The online version of this article includes the following figure supplement(s) for figure 1:

**Figure supplement 1.** The automated and experimenter-free home cage and touchscreen testing system.

**Figure supplement 2.** Schematic overview of touchscreen pre-training and serial reversal learning protocol.

prior to behavioral intervention. Food (standard laboratory rodent chow; Barastoc Feeds, AU) was provided daily prior to the dark phase throughout the duration of the experiment to maintain ~90% of free-feeding body weight. Because of the young age of the animals, this 90% was increased each week by 10% to account for the normal growth curve during development (Charles River Laboratories). The system was housed in a temperature (22–24°C) and humidity (30–50%) controlled room under a reversed 12 hr light/dark cycle (lights off at 1200 hr).

The custom-designed home cage (26 cm × 34 cm × 55 cm) was placed above an array of twenty RFID readers to track the movement of rats. An automated sorter cage connected the home cage to the testing chamber via two plastic tunnels (8.5 cm in diameter). The automated sorting system consisted of a sorter cage which was positioned directly above a scale for body weight recording, two RFID readers for animal identification, and between two software-controlled gates. Selective passage of a single rat from the home cage to the testing chamber required RFID detection and matching

recorded body weight with pre-set weight defined within the PhenoSoft software. The trapezoid testing chamber consisted of two walls, a touchscreen and on the opposing wall a food magazine with an LED light. A test-specific touchscreen mask with windows at the top and bottom that was dependent on the testing/training phase was placed in front of the touchscreen. The touchscreen is illuminated with white light through the windows at the top of the mask to act as a house light to signal incorrect responses. Sucrose pellets (2.5 mm; Homeopathic Supply Company, UK) were used as rewards and delivered from an automated pellet dispenser positioned outside the testing chamber into the food magazine. Touches to the screen and the delivery and collection of food rewards were detected by the breaking of infrared (IR) beams. Conditioned stimuli consisted of a positive (high) tone and the illumination of LED light within the food magazine. Incorrect or omitted responses resulted in a negative (low) tone and the house light mentioned above, followed by a 'time out' period. Rats were allowed to return to the home cage via the sorter following the completion of cognitive tests, which were operated by the PhenoSoft program (PhenoSys, Berlin).

## Pre-training to shape reward-based behaviors

A series of pre-training stages including Habituation, Initial Touch, Must Touch, Must Initiate, and Punish Incorrect were used to shape reward-based behaviors of the rats toward the touchscreen (see *Figure 1—figure supplement 2* & *Supplementary file 1* for details). A mask with three side-by-side windows was used in all pre-training stages. Rats were allowed to have multiple sessions of training per day, with a maximum duration of 30 min or 30 trials per session and a 1 hr time-out period between sessions.

## Pairwise discrimination and reversal learning

The pairwise discrimination (PD) and reversal learning (RL) task were used to assess cognitive flexibility in rats. A touchscreen mask with two side-by-side windows was used in the task. Rats were allowed to perform multiple sessions per day, as in pre-training stages, and were first required to discriminate between two stimulus images (*Figure 1—figure supplement 2E*) and associate touching one of the images with receiving the reward. Rats were required to complete 2 sessions (2 × 30 positive trials) with accuracy >80% within one day to reach the progression criterion to reversal learning, in which the stimulus-reward association was reversed. The progression criterion in reversal learning remained the same as in pairwise discrimination. The training and testing protocols were adapted from previous studies (*Mar et al., 2013*; *Oomen et al., 2013*) with modifications to accommodate the automated system (see *Figure 1—figure supplement 2* & *Supplementary file 1* for details). To assess ABA and flexible learning in the same animals, each rat was restricted to a maximum of 20 sessions of reversal learning to prevent touchscreen overtraining. Once rats reached either the progression criterion (i.e. learned the task) or 20 sessions of reversal learning (i.e. did not learn the task), they were either transferred to the running wheel cages to undergo the ABA paradigm or removed from the experiments and euthanized with 300 mg/kg sodium pentobarbitone (Lethabarb; Virbac, Australia).

## Activity-based anorexia (ABA)

The ABA paradigm used in this experiment consisted of unlimited access to a running wheel and time-restricted food access. At seven weeks of age, or after reaching the progression criterion of reversal learning, rats were individually housed in transparent living chambers with a removable food basket and a running wheel (Lafayette Instruments, IN, USA) in a temperature (22–24°C) and humidity (30–50%) controlled room under a reversed 12 hr light/dark cycle (lights off at 1100 hr). Rats were allowed to habituate to the living chamber with ad libitum food access and locked wheels for 3 days, then habituated to the running wheel for seven days to determine baseline running wheel activity. Running activity was recorded by the Scurry Activity Wheel Software (Lafayette Instruments, IN, USA). During ABA, food access was restricted to 90 min per day at the onset of the dark phase (1100–1230 hr). Running activity in the hour before the feeding window (1000–1100 hr) was considered as food anticipatory activity. Time-restricted food access persisted for a maximum of 10 days or until rats reached <80% of baseline body weight (ABA criterion). Food-restricted control rats were individually housed in standard cages for 10 days with ad libitum food access, followed by 10 days of time-limited access to food for the same 90 min period as ABA rats. Rats were then allowed to recover to baseline

body weight before progression to subsequent cognitive testing or removal from the experiment and euthanised with 300 mg/kg sodium pentobarbitone (Lethabarb; Virbac, Australia).

## Machine learning tools for tracking movements

To track the body parts of rats over time, FFMPEG (*FFMPEG contributors, 2023*) was used to pre-process the videos before analysis, and videos from the touchscreen chamber were imported into DeepLabCut (*Mathis et al., 2018*; *Nath et al., 2019*; *Mathis et al., 2022*). One experimenter labeled 1182 frames from nine videos with the most variation in camera lighting. We trained a ResNet-50 neural network (*Insafutdinov et al., 2016*; *He et al., 2016*) for 200,000 iterations using a training fraction of 80%. We used 1 shuffle and the errors for test and training were 3.97 pixels and 3.13 pixels, respectively. For comparison, the image sizes were 576 by 432 pixels. Data was imported into DLCAnalyzer and all DeepLabCut predictions for the nose point were smoothed using a median filter with a window duration of 0.17 s (five frames). All default settings were used except a global scale of 1 and a p-cutoff of 0.

In order to explore possible differences in behavior that may contribute to cognitive performance in touchscreen-based tasks, DeepLabCut-tracking data was imported into B-SOiD(*Hsu and Yttri, 2021*, *Hsu and Yttri, 2023*). The tracking data for the nose point, left ear, right ear, left hip, right hip, and tail base was used to train an unsupervised behavioral segmentation model. The video frame rate was selected as 30 fps. We randomly selected 49% of the data and B-SOiD randomly subsampled 12% of that data (input training fraction of 0.12). The minimum time length for clusters to exist within the training data was adjusted to yield 34 clusters (cluster range of 0.17–2.5%). These 34 clusters were manually grouped into six behaviors by interpreting video snippets of behaviors that last >300 ms. These behaviors are grooming, inactive, investigating (nose interacts with either the pellet dispenser or images), locomote (walking forwards), rearing, and rotating body. Fleeting behavioral bouts that lasted <300 ms were also replaced with the last known behavior.

The codes used for each of these steps can be found here, (copy archived at *Dempsey et al., 2022b*). This includes all the codes and example data needed to reproduce this analysis from the touchscreen chamber videos to the spider plots and time bin heatmaps.

## Exclusions

To assess whether cognitive flexibility predicted susceptibility to weight loss in ABA, rats that failed to reach the progression criterion within 20 sessions of reversal learning were excluded from all behavioral and performance data analyses because their levels of flexible learning were unable to be assigned (n=3). In addition, three rats demonstrated abnormal weight loss trajectory due to food hoarding in ABA and were therefore excluded from all ABA analyses, as this behavior confounds the generation of the ABA phenotype. Moreover, one rat failed to recover to >80% baseline body weight following exposure to ABA and one rat failed to learn pre-training to shape reward-based behavior towards the touchscreen after prior exposure to ABA. These two animals were excluded from all data analyses, resulting in a final sample size of n=22 in the assessment of the effect of prior exposure to ABA on cognitive flexibility. All sessions post-criterion or with technical issues were excluded from performance and behavioral analyses.

## Data processing and statistical analyses

Daily data output files from the touchscreen, sorter, and activity monitor were processed using our freely available python-based data analysis pipeline to provide detailed information about the performance of each rat during their touchscreen sessions. Statistical analyses were performed using GraphPad Prism 9.1.1 (GraphPad Software, San Diego, CA, USA). Statistical significance was considered as $p < 0.05$ and analyses including Log-rank (Mantel-Cox) test, two-tailed unpaired t-test, linear regression, correlation, one-way and two-way analysis of variance (ANOVA) with Tukey's or Bonferroni's post hoc multiple comparisons were used according to the number of groups and type of data. With respect to performance measures in the touchscreen task, the number of trials to performance criteria together with the trial type are the most informative, because they relate directly to the number of action-outcome associations required for learning. However, the number of sessions required to reach performance criteria is also important for understanding the speed of learning and consolidation. We

therefore report both (trials and session) in the analysis of learning outcomes. Full details of statistical tests performed in these studies can be found in the **Source data files** accompanying each figure.

## Results

### System validation & optimization

Prior to experiments involving the ABA model, we first conducted a series of experiments to validate and optimize the use of the novel testing system in young female rats. We demonstrated that the automated system reduces the number of testing days to reach the reversal learning performance criterion by up to 10 times compared to conventional touchscreen testing methods (i.e. *Milton et al., 2021*; *Figure 2A*). We also confirmed that subsequent reversals were progressively easier to learn than the initial reversal with fewer sessions required to reach the performance criterion (first reversal > second reversal: p=0.0099; first reversal > third reversal: p=0.0070; *Figure 2B*). Interestingly, the speed of learning serial reversals was driven largely by reduced omissions at the second and third reversals (all ps <0.0309; *Figure 2D–E*). One plausible contributor to the high number of omitted trials is the time of day, because animals can initiate sessions when they are motivated to perform the task as well as if they are simply exploring the touchscreen chamber. Considering that laboratory rats are well-known to be more active in the dark phase, we compared performance between animals who retained unlimited touchscreen access to those that had access restricted to the dark phase (*Figure 2F–K*). Restricting access to the dark phase increased accuracy overall (p=0.0039; *Figure 2G*) and especially in pairwise discrimination (p=0.0371), which was specific for initial learning, whereby more substantial between-group differences were seen during the first 100 trials (p=0.0030; *Figure 2H*). Dark-phase restriction also reduced the number of omitted responses during both pairwise discrimination and reversal (*Figure 2I*), however, this was not significantly different overall (p=0.0737; *Figure 2J*) but rather restricted to the initial stages of discrimination and reversal learning (pairwise discrimination p=0.0024; reversal p=0.0332; *Figure 2K*). The reduced variability in responding within the restricted access group throughout serial reversal learning (see *Figure 2F & I*) is likely to be driven by an increase in motivation that is facilitated by restricted access, and although the time of day did not appear to systematically alter performance in animals with unrestricted access (all trials p=0.2088; trials responded to p=0.1766; *Figure 2—figure supplement 1*), we adopted this dark phase restricted approach for subsequent experimental cohorts. Importantly, none of our experimental groups differed in their rate of acquisition of the pretraining stages of the touchscreen task (*Figure 2—figure supplement 2*), ruling out broad-spectrum effects of ABA exposure and susceptibility on visual operant learning.

### Reciprocal interactions between ABA exposure and cognitive flexibility

In order to determine whether individual differences in flexible learning could *predict* susceptibility to pathological weight loss in ABA, we tested animals on the reversal learning task prior to exposure to ABA conditions (*Figure 3A*). Our previous ABA studies demonstrate that rats segregate into susceptible and resistant subpopulations and in the present study, 12/22 (55%) rats exposed to ABA conditions were able to maintain body weight above 80% of baseline throughout the 10 days of ABA, therefore being classified as 'resistant.' This allowed us to retrospectively compare reversal learning between groups to assess predisposing factors to pathological weight loss. Susceptible and resistant rats differed on all key ABA parameters (i.e. body weight loss trajectory, food intake, running activity) as we have previously published (*Milton et al., 2018*; *Milton et al., 2022*) (see *Figure 3—figure supplement 1*), and resistant rats also spent less time moving than susceptible rats during touchscreen sessions (see *Figure 3—figure supplement 3*). Rats that went on to be susceptible to ABA were able to learn pairwise discrimination at the same rate as rats that went on to be resistant to ABA, as demonstrated by a similar number of sessions and trials to reach the performance criterion (all ps>0.8214; *Figure 3B–D*). Interestingly, rats that went on to be *resistant* to weight loss in ABA required significantly more sessions at the first reversal to reach the performance criterion (p=0.0142; *Figure 3B*), and although this did not translate to a significant increase in overall trials required (p=0.1132; *Figure 3C*) it related specifically to an increase in non-correct responses (i.e. incorrect + omitted responses, p=0.0401; *Figure 3E*), a finding that was reinforced by examining learning rate in the early perseverative phase of reversal (first 100 trials; *Figure 3F*).

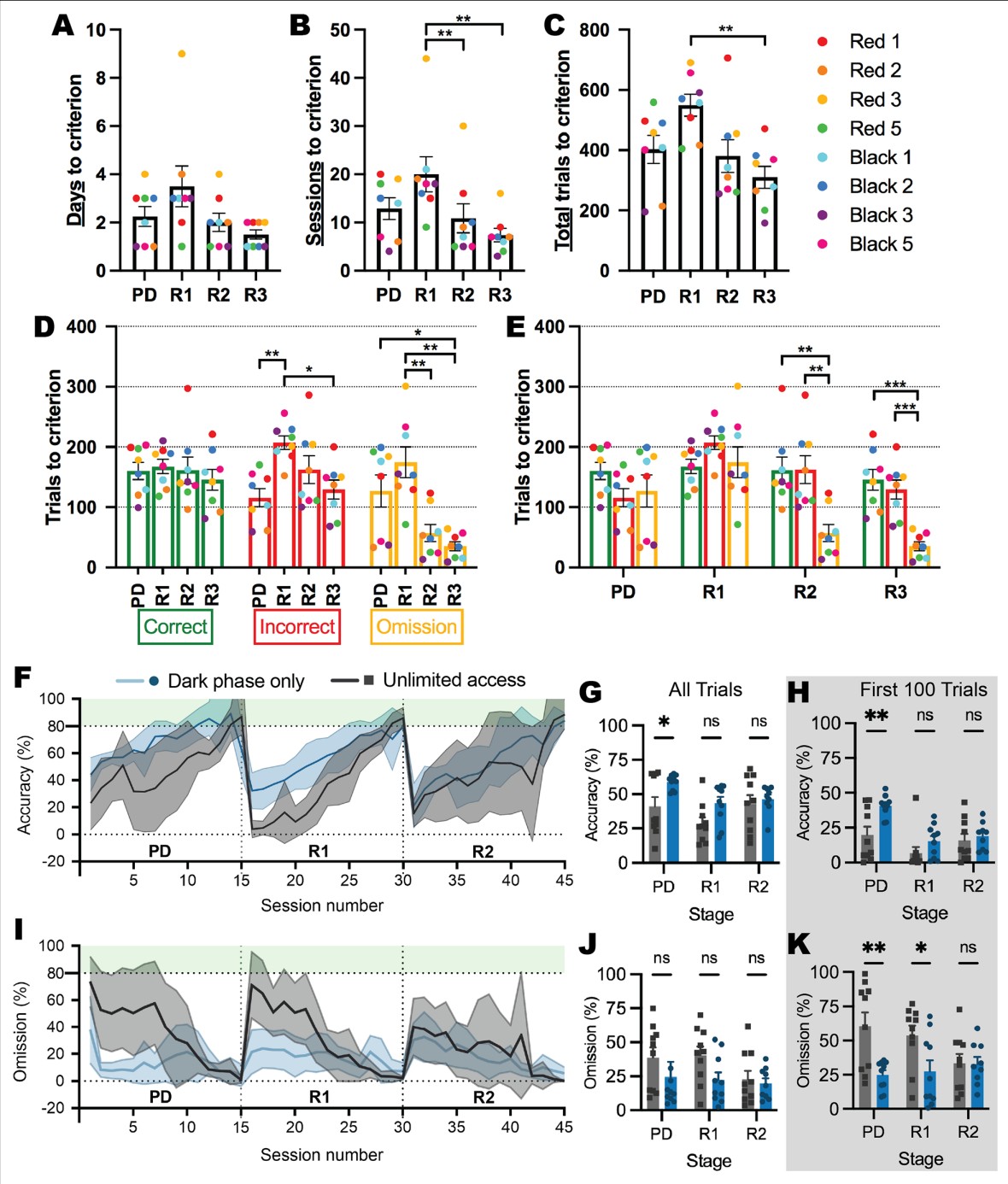

**Figure 2.** Learning rate over serial reversals and effects of unlimited versus dark-phase only access to touchscreen chambers. (**A–E**) Response types over pairwise discrimination (PD), first reversal (R1), second reversal (R2),and third reversal (R3) with unlimited touchscreen access. (**A**) Number of days to criterion (p=0.0484). (**B**) Number of sessions to criterion (p=0.0092): R1>R2 (p=0.0099), R1>R3 (p=0.0070). (**C**) Number of total trials to criterion (p=0.0034): R1>R3 (p=0.0035). Outcome of trials to criterion grouped by outcome (**D**; p=0.0032) and by phase (**E**; p<0.0001). (**D**) Incorrect: R1>PD (p=0.0014), R1>R3 (p=0.0309); Omission: PD>R3 (p=0.0484), R1>R2 (p=0.0092), R1>R3 (p=0.0018). (**E**) R2: correct>omission (p=0.0045), incorrect>omission (p=0.0059); R3: correct>omission (p=0.0005), incorrect>omission (p=0.0008). (**F–K**) Effects of unlimited versus dark-phase-only access on cognitive performance. (**F**) Percentage of correct trials across 15 sessions of each phase of the experiment. (**G**) Percentage of correct trials (access p=0.0039): PD: Dark phase only>unlimited access (p=0.0371). (**H**) Percentage of correct trials in the first 100 trials (access p=0.0030): PD: Dark phase only>unlimited access (p=0.0030). (**I**) Percentage of omission trials across 15 sessions of each phase of the experiment. (**J**) Percentage of omission trials (access p=0.0737). (**K**) Percentage of omission trials in the first 100 trials (access p=0.0008): PD: Dark phase only>unlimited access (p=0.0024); R1: Dark

*Figure 2 continued on next page*

*Figure 2 continued*

phase only>unlimited access (p=0.0332). Bar graphs show group mean ± SEM with individual animals (symbols). Line graphs show group mean ± SEM (shaded bands). *p<0.05, **p<0.01, ***p<0.001.

The online version of this article includes the following source data and figure supplement(s) for figure 2:

**Source data 1.** Full statistical analysis details and results for *Figure 2* and supplements.

**Figure supplement 1.** Time of day does not influence PhenoSys touchscreen performance.

**Figure supplement 2.** Touchscreen pre-training performance measures.

To investigate the behavioral correlates of cognitive task performance that might differentiate rats susceptible versus resistant to weight loss in ABA, we used the DeepLabCut and B-SOiD machine learning tools to annotate videos from touchscreen sessions that were used to train a prediction model, and clustered behaviours based on this model. Analysis of behavioral profiles during touchscreen testing sessions revealed that during initial discrimination learning, rats that went on to be resistant to ABA spent more time engaged in vertical exploration (rearing; p=0.0336) and locomoting (p=0.0190) compared to susceptible rats, which were significantly more inactive (p<0.0001) during touchscreen testing sessions (*Figure 3—figure supplement 3A*). This differential behavioral profile was similar for reversal learning sessions, with increased rearing again evident in rats that would go on to be resistant to ABA (p=0.0384) and increased inactive time for susceptible rats (p<0.0001), suggesting a consistent exploratory difference between groups even prior to ABA exposure (*Figure 3—figure supplement 3B*) that may contribute to variation in susceptibility to weight loss.

To examine whether prior exposure to ABA conditions elicited a persistent change in cognitive flexibility, we allowed animals to recover their body weight to >100% of pre-exposure levels before testing them in the automated touchscreen system (*Figure 4A*). Here, we show that ABA produced a profound impairment in both discrimination and flexible learning, even after weight recovery, that did not occur to the same extent following a matched period of food restriction alone. Not only were half (50%) of ABA-exposed animals unable to acquire the reversal learning task, compared to 32% of animals exposed to restricted food access and 11% of ABA-naïve animals (*Figure 4B*), task acquisition was significantly slower than ABA-naïve rats. Exposure to ABA conditions increased the number of sessions required to reach performance criteria compared to ABA-naïve animals (ABA exposure p=0.0051; pairwise discrimination p=0.0098; reversal p=0.0205) with a similar trend compared to food restriction-only animals (ABA exposure p=0.0858; pairwise discrimination p=0.0539; reversal p=0.3985; *Figure 4C*). In contrast, the ABA-naïve and food restriction-only groups were not different (all ps>0.6758). While there was no overall difference in the number of trials required to reach progression criteria across the entirety of the experiment (p=0.1240; *Figure 4D*), the number of correct (versus ABA-naïve p=0.0185; versus food restriction-only p=0.0259), incorrect (versus food restriction-only p=0.0479) and omitted (versus ABA-naïve p=0.0224) trials to the acquisition of initial discrimination were significantly greater in the ABA-exposed group (*Figure 4E*). Importantly, there were no differences in the number of trials to performance criterion required for ABA-naïve and food restricted-only control animals to learn pairwise discrimination (all ps>0.9999; *Figure 4E*). While the number of total trials and of each response type required to learn the reversed contingencies did not differ between animals that were able to learn the reversal task (all ps>0.2656; *Figure 4F*), this result is confounded by the large proportion of ABA-exposed animals that did not reach performance criterion for the reversal phase of the task. The specific impairment in reversal learning elicited by ABA exposure is evident in the trials per session data displayed in *Figure 4—figure supplement 1* (and described below).

When considering the response profiles of both the ABA-exposed and food restriction-only animals that were unable to learn the reversal task, it was clear that this was not related to impaired performance on aspects of discrimination learning, with similar numbers of sessions (p=0.8445 and p=0.4656 respectively; *Figure 4G* and *Figure 4—figure supplement 1J*) and trials (p=0.5626 and p=0.6133 respectively; *Figure 4H* and *Figure 4—figure supplement 1K*) required to reach performance criterion compared to animals within each group that were able to learn. The types of trials required for animals that did and did not learn the reversal task in these two groups were also unchanged for discrimination learning (all ps>.3892 and all ps>0.6133; *Figure 4I* and *Figure 4—figure supplement 1L*), however, within each group both the number of correct (both ps<*0.0001; Figure 4—figure*

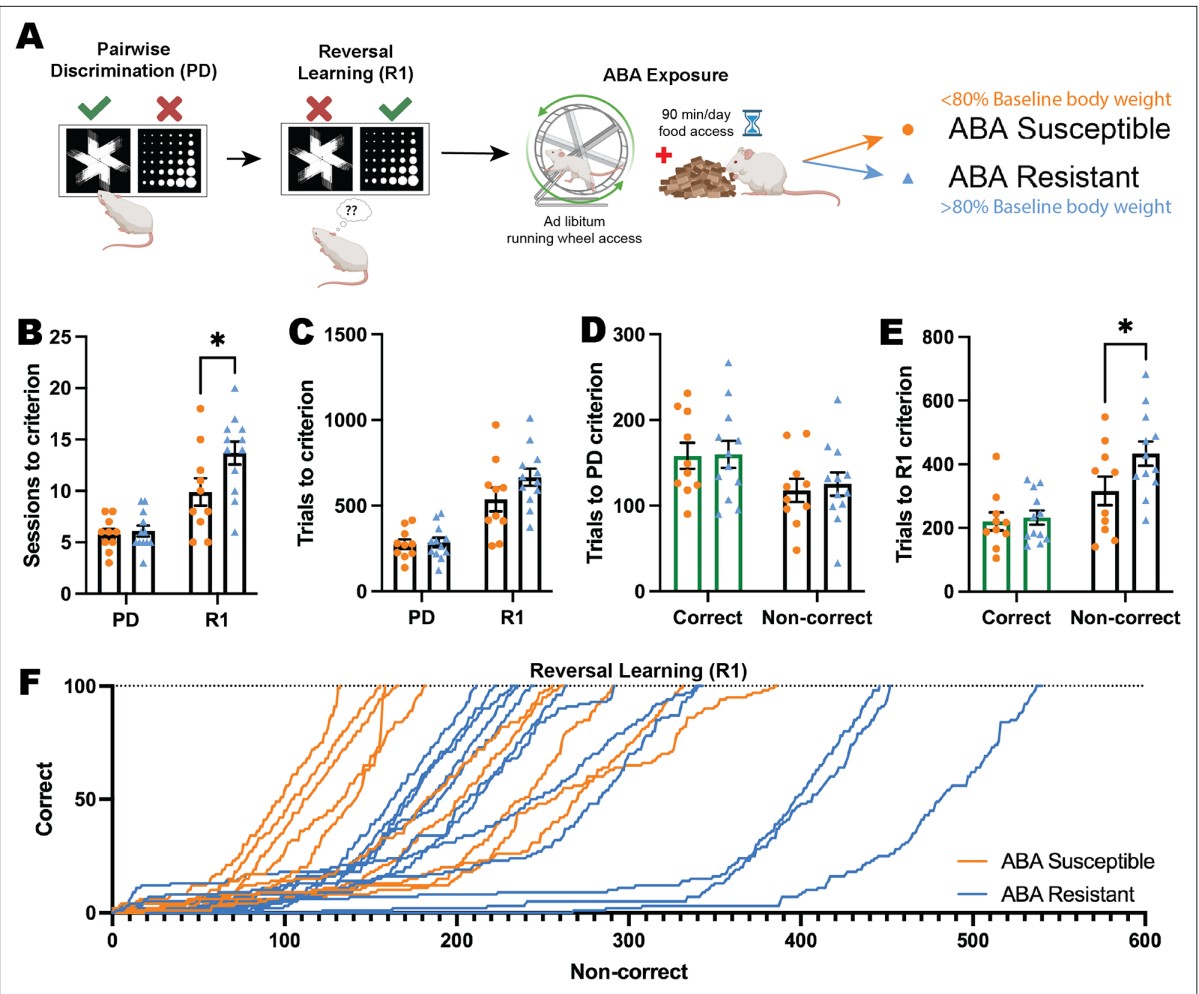

**Figure 3.** Impaired cognitive flexibility does not predict susceptibility to ABA. (**A**) Schematic of pairwise discrimination (PD) and reversal learning (R1) task and subsequent activity-based anorexia paradigm (ABA). Animals split into two experimental groups determined by body weight loss after exposure to ABA: susceptible or resistant to ABA. Bar graphs show group mean ± SEM with individual animals (symbols). (**B**) Number of sessions to reach criterion (outcome*phase interaction p=0.0292): R1: ABA resistant>ABA susceptible (p=0.0142). (**C**) Number of total trials to reach criterion. (**D**) Number of correct or non-correct trials to reach the PD criterion. (**E**) Number of correct or non-correct trials to reach R1 criterion (outcome*phase interaction p=0.0389): Non-correct trials: ABA resistant>ABA susceptible (p=0.0401). (**F**) Progressive performance across the first correct 100 trials in R1 for individual animals: Non-correct response → X+1; correct response → Y+1. *p<0.05.

The online version of this article includes the following source data and figure supplement(s) for figure 3:

**Source data 1.** Full statistical analysis details and results for *Figure 3* and supplements.

**Figure supplement 1.** Key activity-based anorexia (ABA) parameters that differentiate individuals that are susceptible and resistant to ABA and behavioral profiles during cognitive testing.

**Figure supplement 2.** Age at onset of activity-based anorexia (ABA) following completion of the Reversal Task.

**Figure supplement 3.** Specific behavioral profiles identified by machine learning tools during touchscreen testing could be used to predict susceptibility or resistance to activity-based anorexia (ABA).

*supplement 1M*) and incorrect (ABA exposed p=0.0002; food restriction only p<0.0001; *Figure 4—figure supplement 1N*) trials per session were substantially reduced for 'non-learners' *specifically when reward contingencies were reversed*. Together with the absence of a significant difference in the number of omitted trials per reversal session (both p*s*>0.9999; *Figure 4—figure supplement 1O*), this indicates that a lack of reward-based feedback (either positive or negative) impaired the ability of these subgroups of ABA exposed or food restriction only animals to flexibly update responding in the reversal task. Video analysis of touchscreen sessions revealed that during pairwise discrimination, ABA-exposed animals spent more time inactive and less time engaged in task-relevant behaviors like

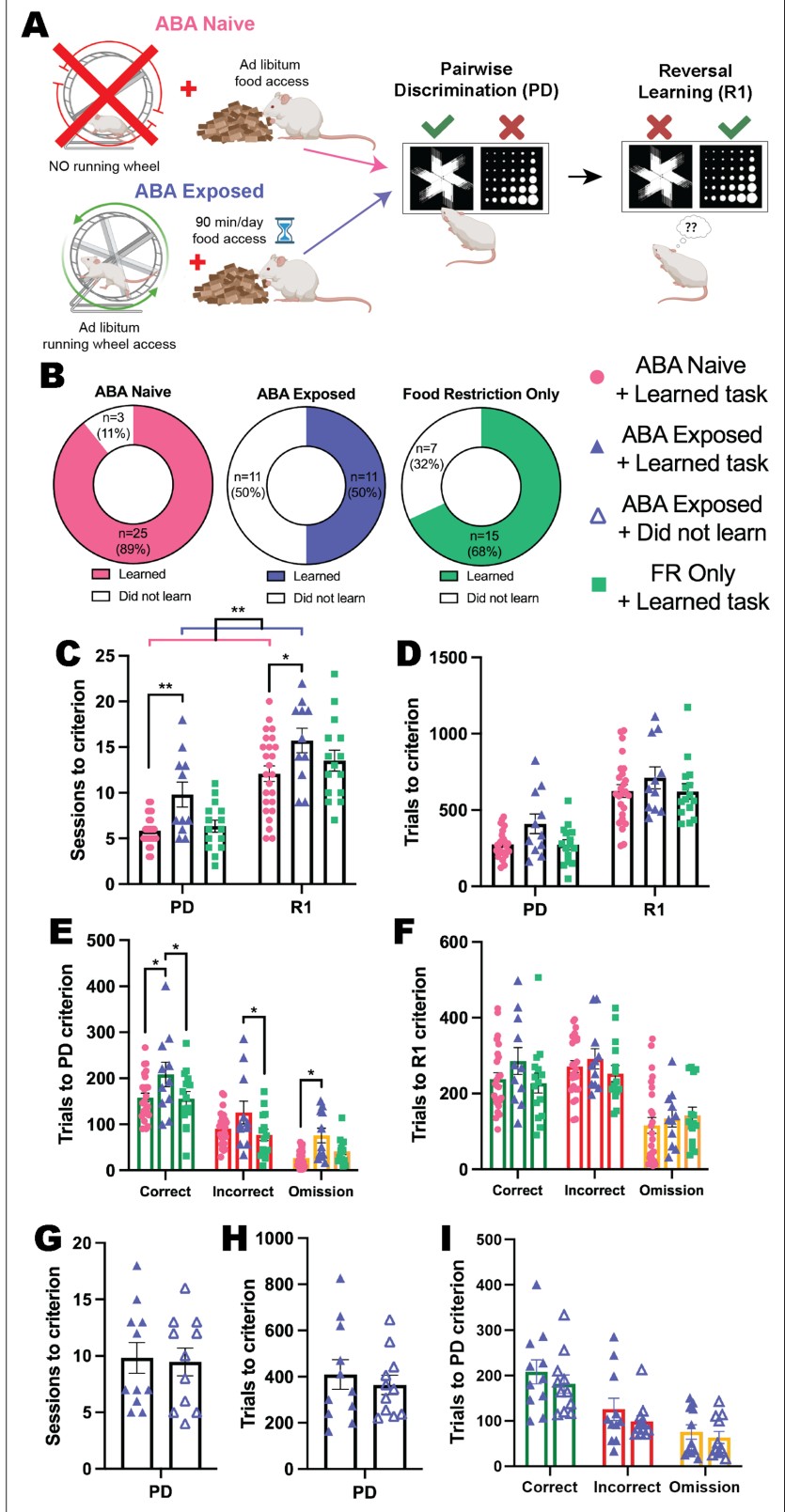

**Figure 4.** Exposure to ABA conditions impairs cognitive performance. (**A**) Schematic of experimental paradigm showing activity-based anorexia (ABA) Naïve or Exposed groups and the subsequent pairwise discrimination (PD) and reversal learning (R1) task. There was also a Food restriction-only group that underwent the same time-restricted feeding as the ABA groups but did not have access to running wheels. Animals were split into six

*Figure 4 continued on next page*

*Figure 4 continued*

experimental groups: Naïve rats that were not exposed to ABA conditions and learned the reversal learning task (ABA Naïve + learned task); ABA Naïve but did not learn the task (ABA Naïve + did not learn); rats previously exposed to ABA condition that learned the subsequent task (ABA Exposed + learned task); rats previously exposed to ABA that did not learn the task (ABA Exposed + did not learn); Food restriction-only rats that learned the task (FR only + learned task); and Food restriction only rats that failed to learn the reversal task (FR only + did not learn). (**B**) Donut plots of experimental groups: 89% (25/28) of the ABA Naïve rats and 68% (15/22) of the food restriction only rats learned the reversal learning task compared to only 50% (11/22) of the ABA Exposed rats. (**C**) Number of sessions to reach criterion (main effect of ABA exposure p=0.0068; ABA Exposed + learned task>ABA Naïve + learned task p=0.0051): PD: ABA Exposed + learned task>ABA Naïve + learned task (p=0.0098); R1: ABA Exposed + learned task>ABA Naïve + learned task (p=0.0205). (**D**) Number of total trials to criterion. (**E**) Number of correct, incorrect, and omission trials to PD criterion (ABA exposure p=0.0181; ABA Exposed + learned task>ABA Naïve + learned task p=0.0241, ABA Exposed + learned task>Food restriction only + learned task p=0.0412). Correct: ABA Exposed + learned task>ABA Naïve + learned task (p=0.0185), ABA Exposed + learned task>Food restriction only + learned task (p=0.0259). Incorrect: ABA Exposed + learned task>Food restriction only + learned task (p=0.0479). Omission: ABA Exposed + learned task>ABA Naïve + learned task (p=0.0224). (**F**) Number of correct, incorrect and omission trials to R1 criterion. Number of (**G**) sessions, (**H**) total trials, and (**I**) correct, incorrect and omission trials to PD criterion. Bar graphs show group mean ± SEM with individual animals (symbols). *p<0.05, **p<0.01.

The online version of this article includes the following source data and figure supplement(s) for figure 4:

**Source data 1.** Full statistical analysis details and results for *Figure 4* and supplements.

**Figure supplement 1.** Comparison of trial types between activity-based anorexia (ABA) naïve rats and ABA exposed or food restriction-only rats that did or did not learn the reversal task.

**Figure supplement 2.** Lone activity-based anorexia (ABA)-exposed Resistant rat highlighted red by request of Reviewers.

**Figure supplement 3.** Behavioral differences during touchscreen testing due to order effects of cognitive task and activity-based anorexia (ABA) exposure.

**Figure supplement 4.** Behavioral differences during touchscreen testing due to whether rats learned or did not learn first reversal after prior exposure to activity-based anorexia (ABA).

**Figure supplement 5.** Performance during the first and last pairwise discrimination (PD) and first reversal (R1) sessions.

rotating, investigating, and magazine interactions than did ABA-naïve animals, whereas behavioral profiles were more similar between groups for reversal sessions (see *Figure 4—figure supplement 3*). The *specific* impairment in reversal performance in the ABA-exposed animals was reflected by more substantial differences between 'non-learners' and 'learners' in time spent inactive during reversal compared to pairwise discrimination sessions, and by the specific reduction in task-relevant activities including interactions with the reward magazine and touchscreen images in reversal sessions only (see *Figure 4—figure supplement 4*).

Finally, to explore whether cognitive testing changed the development of the ABA phenotype, we compared ABA outcomes for touchscreen-testing naïve animals (Before Reversal Task) to those that occurred following touchscreen-based reversal learning (After Reversal Task; *Figure 5A*). Significantly more rats that underwent cognitive testing prior to ABA were able to resist the precipitous weight loss that characterizes the model (p<0.0001; *Figure 5B*) and demonstrated a slow trajectory of body weight loss that plateaued over consecutive days of ABA exposure (*Figure 5C*). When comparing outcomes for both susceptible and resistant animals on key ABA measures, those that had undergone touchscreen testing prior to ABA lost significantly less body weight each day (p<0.0001; *Figure 5D*) and ate more food when food was available (p=0.0009; *Figure 5E*). As expected, rats that only under-went food restriction without access to a running wheel lost significantly less body weight (p<0.0001) and ate more food in the 90 min window (p=0.0045) than ABA-exposed animals (*Figure 5—figure supplement 1*). Rats that had undergone touchscreen testing prior to ABA also showed a blunted hyperactive phenotype when ABA conditions commenced that was already evident under baseline conditions (p<0.0001; *Figure 5F*). Although running activity overall was significantly reduced in animals that had previously undergone cognitive testing both at baseline and during ABA (overall p=0.0002; baseline p=0.0160; ABA p<0.0001; *Figure 5G*), these rats showed elevated running specifically in

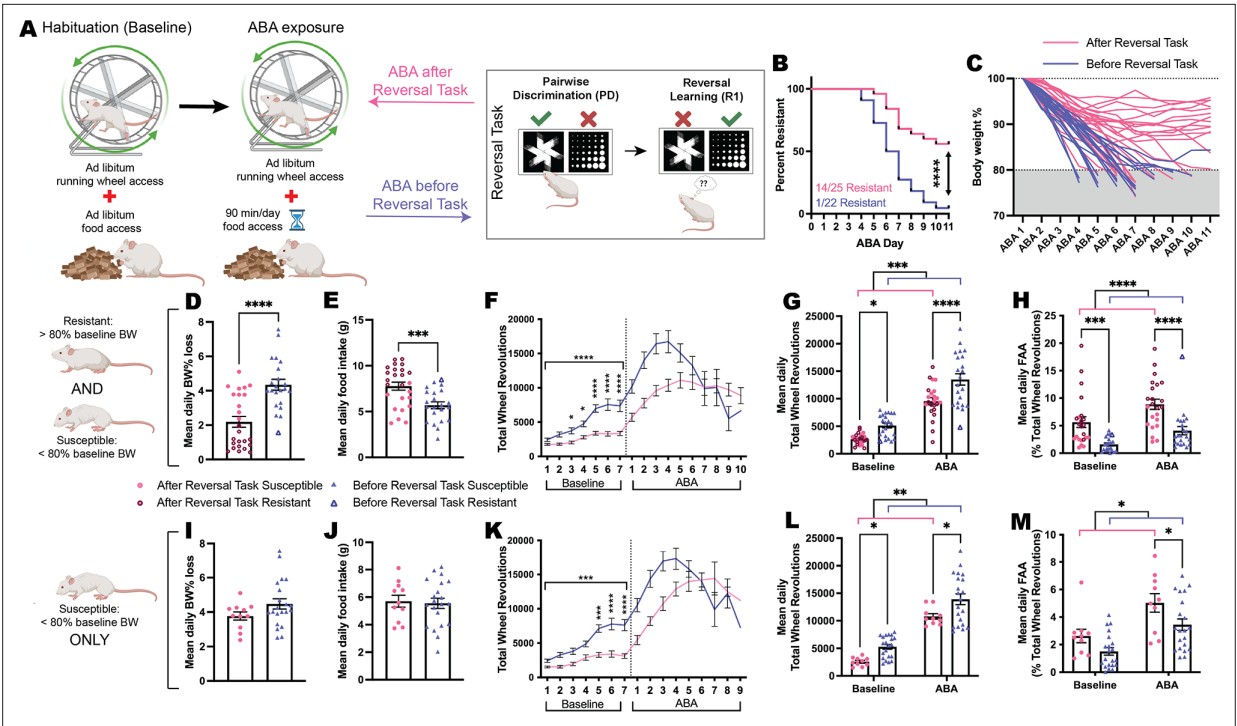

**Figure 5.** Prior training on the reversal task changes the development of the ABA phenotype. (**A**) Schematic of activity-based anorexia (ABA) paradigm and the prior or subsequent pairwise discrimination (PD) and reversal learning (RL) task in Reversal Task. (**B**) Survival plot comparing order effects: ABA resistance was 56% (14/25) for rats that were exposed to ABA <u>after</u> Reversal Task compared to 5% (1/22) for rats that underwent ABA <u>before</u> Reversal Task (p<0.0001). (**C**) Body weight (% of baseline) trajectories for individual animals across a maximum of 10 days of ABA or until they reached <80%. Data shown are from ALL animals that underwent ABA (**D, E, G, H**) or ONLY ABA susceptible animals (**I, J, L, M**). (**D**) Mean daily ABA body weight (BW) % loss, Before Reversal Task>After Reversal Task (p<0.0001). (**E**) Mean daily ABA food intake, After Reversal Task>Before Reversal Task (p=0.0009). (**F**) Daily running wheel activity (RWA) across both experimental phases. <u>Baseline</u>, all ps<0.0001: Before Reversal Task>After Reversal Task (Day 3, p=0.0440; Day 4, p=0.0105; Days 5–7, all ps<0.0001). (**G**) Mean daily RWA (ABA timing p=0.0002): Before Reversal Task>After Reversal Task during both baseline (p=0.0160) and ABA (p<0.0001). (**H**) Mean daily food anticipatory activity (FAA; RWA in the hour before food access; ABA timing p<0.0001). After Reversal Task>Before Reversal Task during both baseline (p=0.0010) and ABA (p<0.0001). Mean daily ABA body weight % loss (**I**) and food intake (**J**). (**K**) Daily RWA across both experimental phases. Baseline, all ps<0.0002: Susceptible Before Reversal Task>Susceptible After Reversal Task (Day 5: p=0.0001; Days 6–7: ps<0.0001). (**L**) Mean daily RWA (ABA timing p=0.0065): Susceptible Before Reversal Task>Susceptible After Reversal Task during both baseline (p=0.0426) and ABA (p=0.0165). (**M**) Mean daily FAA (ABA timing p=0.0157). Susceptible After Reversal Task>Susceptible Before Reversal Task during ABA (p=0.0357). Bar graphs show group mean ± SEM with all individual animals (symbols); line graphs show group mean ± SEM. *p<0.05, **p<0.01, ***p<0.001, ****p<0.0001.

The online version of this article includes the following source data and figure supplement(s) for figure 5:

**Source data 1.** Full statistical analysis details and results for *Figure 5* and supplements.

**Figure supplement 1.** Restricting food access without a running wheel does not lead to significant weight loss.

---

the hour preceding food access, known as food anticipatory activity, which is an adaptive response to scheduled feeding (overall p<0.0001; baseline p=0.0010; ABA p<0.0001; *Figure 5H*). While our previous work has shown elevated food anticipatory activity to be consistently associated with resistance to ABA (*Milton et al., 2021*; *Milton et al., 2018*; *Milton et al., 2022*), the increased food anticipatory activity at baseline for these animals suggests that an anticipatory response was carried over from the scheduled feeding conducted during touchscreen testing. Considering that exposure to cognitive training significantly increased the percentage of rats that were resistant to ABA, it was important to also examine the effects of cognitive training on ABA outcomes in only those rats susceptible to weight loss. The concern was that any differences in ABA outcomes may be driven solely by this subpopulation of resistant animals. Neither mean daily weight loss (p=0.1277; *Figure 5I*) nor food intake (p=0.7794; *Figure 5J*) were differentially altered by prior cognitive testing in susceptible rats, however, there remained significantly reduced levels of running wheel activity in susceptible rats following discrimination and reversal learning (p=0.0002; *Figure 5K*). Again, susceptible rats that

had previously undergone cognitive training ran less both at baseline (p=0.0426) and during ABA conditions (p=0.0165; *Figure 5L*) whereas food anticipatory activity was specifically increased only during ABA (p=0.0357; *Figure 5M*). Taken together, these data suggest that cognitive training alters the development of the ABA phenotype specifically through attenuating excessive running activity.

## Discussion

Here, we validate and optimize the use of a novel automated and experimenter-free touchscreen testing platform for rats and demonstrate the application of this system for rapid assessment of cognitive flexibility before and after exposure to ABA. Critically, the rate of learning in the automated system was shown to be five times faster (with approximately 10 times higher throughput) than previously reported with conventional touchscreen testing (*Milton et al., 2021*). While the full spectrum of possibilities arising from the use of the modular PhenoSys touchscreen system is still being realized, the increased throughput, the requirement for fewer animals, and reduced labor time for experimenters represents a major shift in the way these experiments are conducted and analyzed. Our observation that the number of omitted trials is reduced (i.e. engagement is higher) when touchscreen access was limited to the dark phase is consistent with the well-established increase in activity (*Milton et al., 2021*) and attention (*Bruinsma et al., 2019*) that rats exhibit during the dark period. Moreover, the ability to rapidly test cognitive flexibility with the automated touchscreen system allowed us, for the first time, to examine the cognitive profiles of animals prior to exposure to the ABA paradigm while ensuring that rats remained young adults for ABA exposure. In addition, we conducted this assay in socially appropriate groups and without experimenter intervention, increasing the reliability of outcomes and removing potential confounds of handling on subsequent ABA phenotypes (*Carrera et al., 2006*).

Our previous work demonstrated that activity within a specific neural circuit (extending from the medial prefrontal cortex to the nucleus accumbens shell) links pathological weight loss in ABA with cognitive inflexibility on this reversal learning task (*Milton et al., 2021*) and suggested that inflexibility might be a biomarker for predicting susceptibility to ABA. The results presented here demonstrate that, contrary to our hypothesis, inflexibility does not predispose animals to the ABA phenotype but instead shows that rats that went on to be resistant to ABA were slower to learn the reversal task (i.e. were less flexible) than ABA-susceptible rats. This raises the intriguing possibility that either inflexibility develops coincident with pathological weight loss in the ABA model or that inflexibility in this instance is protective against ABA-induced weight loss. One finding supporting the latter is that rats that went on to be resistant to ABA were hyper-exploratory in touchscreen testing sessions, evidenced by increased rearing behaviors and decreased time spent inactive during the task. Persistently high exploratory activity may result in more errors during touchscreen test sessions as well as hastened food seeking during the feeding window of ABA. Regarding the former, while we did not examine flexible learning *during* exposure to ABA conditions, the idea that inflexibility and ABA develop in concert fits with the timing of neural circuit manipulation used in our previous work (*Milton et al., 2021*). That is, both pathological weight loss and inflexibility were prevented by suppressing the same 'cognitive control' neural circuit, but suppression occurred *during* the development of ABA, not prior to ABA exposure. To this end, future studies should delineate the precise stage during the development of the ABA phenotype where inflexibility becomes apparent, thereby defining a 'therapeutic window' in which novel pharmacological treatments could be tested with greater translational relevance.

It remains the case that the precise mechanisms underlying cognitive inflexibility in individuals with anorexia nervosa are not well understood, and while this reversal learning task may not capture the prodromal inflexibility that might play a role in the pathophysiology of anorexia nervosa, it may still provide important clues to direct us toward the neurobiological underpinnings of aberrant reinforcement learning in patients. For example, if reversal learning is defined as the ability to adapt behavior based on negative feedback (*Robbins et al., 2012*), our demonstration that improved reversal learning is associated with increased susceptibility to ABA fits with the hypothesis that maladaptive behaviors in anorexia nervosa act to alleviate negative affect and are strengthened through heightened negative reinforcement learning (*Coniglio et al., 2022*). Exposure to ABA conditions, but not food restriction alone, significantly impairs cognitive flexibility, indicating that the excessive exercise component of the ABA model disrupts the integration of new and existing learning that is necessary for performance on the reversal learning task.

The development of paradoxical hyperactivity when restricted food access is imposed, a major hallmark of the ABA phenotype, may reflect a compulsive behavior (*Miletta et al., 2020*) defined by persistent wheel running (energy output) despite the negative consequences of rapidly declining body weight. And yet the ability to effectively adapt behavior to environmentally imposed change (reversal learning) was improved in rats that demonstrated the highest levels of food restriction-evoked hyperactivity and became susceptible to ABA. This challenges our conceptualization of the so-called 'compulsive' wheel running that occurs during ABA and precipitates the rapid weight loss characteristic of the model. Even after decades of experimental use of the ABA model, the causes for this paradoxical hyperactivity remain elusive. A recent study in ABA mice demonstrated that a loss of behavioral flexibility following disrupted cholinergic activity in the dorsal striatum was associated with both facilitated habit formation and increased vulnerability to maladaptive eating (*Favier et al., 2020*) but neither the accelerated formation of habits, or inflexible behaviors were associated with changes in hyperactivity. Similarly, although compulsive behavior in individuals with anorexia nervosa has been described to develop under more habitual than goal-directed control (*Foerde et al., 2021*) these associations have been restricted to eating behavior rather than exercise. We hypothesize that excessive exercise in ABA rats (and possibly in individuals with anorexia nervosa) represents not a habitual behavior but rather a form of extreme goal-directed control (*Hogarth, 2020*).

Compulsions have been defined as the repeated, goal-directed selection of a habit (*Bradfield et al., 2017*), and perhaps it is the failure to integrate goal-directed (excessive exercise) and habitual (restrictive feeding) action control that underlies the paradoxical behaviors observed in both human anorexia and the ABA model. Recent evidence suggests that only a subset of patients suffering from anorexia nervosa show enhanced habit formation (*Favier et al., 2020*), which can be differentiated from a more goal-directed patient subgroup by decreased medial orbitofrontal cortex activation during reward anticipation (*Steding et al., 2019*). Together, these data highlight the need to directly examine the reciprocal effects of ABA and goal-directed action; specifically focusing on the ability to adapt action selection after outcome-devaluation. Since the cholinergic function of the dorsal striatum is associated with both maladaptive eating in mice (*Favier et al., 2020*) and the interlacing of new and existing learning (*Bradfield et al., 2013*), the ability to adapt behavior in response to changing outcome values after the reversal of action-outcome pairings (*Matamales et al., 2016*) may be of more relevance to ABA than the reversal of stimulus pairings examined in this study. Understanding how goal-directed and habitual control of wheel running might change over the course of ABA exposure could inform modifications to cognitive-behavioral therapy for individuals with anorexia nervosa based on a perspective of eating disorder-relevant goals, particularly in those ~80% of patients that engage in excessive exercise (*Davis et al., 1997*). Combining the ABA model with cognitive behavioral assays that contrast habitual with goal-directed behavior, including outcome devaluation tasks (*Watson et al., 2022*; *Turner et al., 2022*), and probing the involvement of the dorsal striatum and orbitofrontal cortex could aid substantially in this understanding.

Our data also suggest that operant training prior to exposure to ABA also alters the subsequent development of anorectic phenotypes, particularly by reducing the maladaptive wheel running that typifies the ABA model. This effect is likely to relate to the procedural aspects of training (i.e. mild food restriction combined with sucrose rewards) rather than to task performance, considering that rats that went onto be susceptible to ABA acquired the reversal task more quickly than rats that went onto be resistant. It should also be noted that the cohort of rats that were experimentally naïve prior to ABA exposure was surprisingly mostly susceptible to weight loss (21/22 rats), a much higher proportion than is normally seen on a cohort-to-cohort basis. This single 'resistant' rat was among the fastest ABA-exposed animals to acquire performance criteria in the reversal task (see *Figure 4—figure supplement 2*), but was not a significant group outlier. While the independent effects of sucrose consumption and scheduled feeding on subsequent weight loss in ABA are yet to be determined, if ABA indeed develops through a failure to effectively adapt to the change in food availability, then our results support the idea that *experience* with reinforcement learning tasks (regardless of the speed of task acquisition) improves this adaptive capacity. Interestingly, this aligns with recent evidence that increased cognitive flexibility mediates improvements in eating disorder symptoms in patients with anorexia nervosa (*Duriez et al., 2021*).

While the identification of a behavioral predictor (or biomarker) for pathological weight loss in ABA remains a challenge, the finding that rats exposed to ABA subsequently showed marked impairments

in both discrimination and reversal learning, even after body weight recovery, is entirely in line with the clinical presentation of inflexibility in patients with anorexia nervosa long after body weight recovery (*Tchanturia et al., 2012*; *Friederich and Herzog, 2011*; *Tchanturia et al., 2004*). The finding that exposure to food restriction, resulting in mild weight loss and a non-significant decrease in learning performance, indicates that the severe cognitive impairment seen in ABA-exposed rats was related to the *speed and extent* of weight loss. Intriguingly, this learning impairment induced by ABA exposure was evident from the first session of each phase of training and even within the first 10 min of initial pairwise discrimination performance (see *Figure 4—figure supplement 5*). This lends weight to the use of the ABA model as an effective tool with which to probe the biological mechanisms underlying cognitive deficits in anorexia nervosa. Our finding is in contrast to the only other published report of flexible learning after exposure to ABA (*Allen et al., 2017*), in which reversal learning was impaired at low body weight in ABA rats but ameliorated with weight recovery. Although the reasons for this discrepancy remain unclear, the touchscreen testing system used in the present study differs on multiple procedural levels from the attentional set-shifting task previously used to examine flexible learning, and our results suggest that the visual reversal learning task may be preferable for delineating the lasting effects of ABA exposure on cognitive function. That we observed impairments following ABA not only on flexible updating of operant responses but also initial discrimination learning points to a potential motivational deficit induced by ABA. This is in line with our previous work demonstrating a role for the mesolimbic dopamine circuitry in the development and maintenance of the ABA phenotype (*Foldi et al., 2017*). Considering that exercise behavior in anorexia nervosa is also linked with dopaminergic activity (*Gorrell et al., 2020*), future studies should define the time course over which motivation or reward-based deficits arise during ABA and the specific influence of ABA on dopamine signaling in response to reward anticipation and receipt using in vivo fiber photometric recordings paired with detection of dopamine release (using the GRAB_DA sensor; *Sun et al., 2018*) or dopamine binding (using the dLight sensor *Patriarchi et al., 2018*).

Not only does the automated touchscreen testing system described here allow us to identify cognitive profiles that more accurately reflect the naturalistic behavior of animals, but the incorporation of behavioral segmentation using machine learning also assisted with reducing experimenter biases that are commonly found with manual behavioral scoring. The application of DeepLabCut and B-SOiD to the prediction of behaviors in the present study has allowed for additional exploration of behaviors that could contribute toward cognitive task performance in rats that will aid in the generation of hypotheses to be tested in future studies. Using these tools also enabled the scoring of very large datasets, such as the 185 hr of footage analyzed here. Incorporating this type of analysis with animal models that mimic specific aspects of human pathologies will take us closer than ever before to the identification of biological predictors of pathological weight loss in ABA that could be used in the early detection of anorexia nervosa in at-risk individuals.

## Acknowledgements

The authors acknowledge the incredible technical support from Dr. Karsten Krepinsky (PhenoSys, GmbH; Berlin) and the use of https://www.biorender.com/ in the generation of some figures. We are grateful for financial support for these studies from the Rebecca L Cooper Medical Research Foundation (Project Grant PG2019373-Foldi) and the National Health and Medical Research Council of Australia (Ideas Grant GNT2001722-Foldi).

## Additional information

### Funding

| Funder | Grant reference number | Author |
| --- | --- | --- |
| Rebecca L. Cooper Medical Research Foundation | PG2019373-Foldi | Claire J Foldi |
| National Health and Medical Research Council | GNT2001722-Foldi | Claire J Foldi |

| Funder | Grant reference number | Author |
|--------|------------------------|--------|

The funders had no role in study design, data collection and interpretation, or the decision to submit the work for publication.

## Author contributions
Kaixin Huang, Data curation, Formal analysis, Investigation, Writing – original draft, Writing – review and editing; Laura K Milton, Data curation, Formal analysis, Supervision, Investigation, Methodology, Writing – review and editing; Harry Dempsey, Data curation, Formal analysis, Validation, Investigation, Visualization, Methodology, Writing – review and editing; Stephen J Power, Data curation, Investigation, Writing – original draft; Kyna-Anne Conn, Supervision, Methodology, Writing – review and editing; Zane B Andrews, Supervision, Investigation, Project administration, Writing – review and editing; Claire J Foldi, Conceptualization, Resources, Data curation, Supervision, Funding acquisition, Investigation, Methodology, Writing – original draft, Project administration, Writing – review and editing

## Author ORCIDs
Kaixin Huang http://orcid.org/0000-0002-9746-7947
Laura K Milton http://orcid.org/0009-0007-1664-8337
Harry Dempsey http://orcid.org/0000-0001-5117-6995
Kyna-Anne Conn http://orcid.org/0000-0003-2244-7885
Zane B Andrews http://orcid.org/0000-0002-9097-7944
Claire J Foldi http://orcid.org/0000-0002-3293-8242

## Ethics
All experimental procedures were conducted in accordance with the Australian Code for the care and use of animals for scientific purposes and approved by the Monash Animal Resource Platform Ethics Committee (ERM 29143 and 15171).

## Decision letter and Author response
Decision letter https://doi.org/10.7554/eLife.84961.sa1
Author response https://doi.org/10.7554/eLife.84961.sa2

# Additional files

## Supplementary files
• MDAR checklist

• Supplementary file 1. Touchscreen parameters. Parameters for pretraining and 2VDLR in the novel touchscreen apparatus.

## Data availability
The data generated in this paper can be found at https://doi.org/10.6084/m9.figshare.21539685. A data analysis pipeline for providing the key data per session can be found at https://github.com/Foldi-Lab/PhenoSys-codes, (copy archived at *Dempsey et al., 2022a*). The codes used to create the pose estimation and behavioural segmentation analysis and figures can also be found at https://github.com/Foldi-Lab/PhenoSys-data, (copy archived at *Dempsey et al., 2022b*).

The following dataset was generated:

| Author(s) | Year | Dataset title | Dataset URL | Database and Identifier |
|-----------|------|---------------|-------------|-------------------------|
| Dempsey H | 2023 | Raw data for figures | https://doi.org/10.6084/m9.figshare.21539685 | figshare, 10.6084/m9.figshare.21539685.v2 |

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
