## [Editor Report]

This important manuscript describes a fully automated touchscreen cognitive testing system for rats that reduces the length of training required to learn a task and eliminates the need for daily handling. These features make it possible to assess cognitive behaviors in conjunction with other neurobehavioral paradigms during adolescence, an important advance in the field. The data are compelling in showing that cognitive flexibility does not promote susceptibility to severe weight loss in the activity-based anorexia (ABA) paradigm and support the claim that the cognitive deficits seen in ABA-exposed rats reflect an important clinical feature of the pathophysiology of anorexia nervosa.

---

## [Decision Letter]

**Decision letter after peer review:**

Thank you for submitting your article "Rapid, automated and experimenter-free assessment of cognitive flexibility reveals learning impairments following recovery from activity-based anorexia in female rats" for consideration by *eLife*. Your article has been reviewed by 3 peer reviewers, including Laura A Bradfield as the Reviewing Editor and Reviewer #1, and the evaluation has been overseen by Kate Wassum as the Senior Editor.

The reviewers agree that the paper is novel and provides a thorough assessment of a new, experimenter-free methodology that can be used to assess cognitive deficits as applied to a potential model of anorexia nervosa. However, some revisions are necessary, in particular, to support the claim that the ABA model can be used to model the cognitive inflexibility that develops in the pathophysiological progress of anorexia nervosa.

Essential revisions:

1) We request the addition of a control group with restricted access to food in order to validate that this model can be used to study anorexia. Even if the data argue against the validity of the ABA model to study cognitive inflexibility in anorexia nervosa, the manuscript would still be valuable a) As a methods paper (that includes a detailed discussion and guidance regarding parameters that are most informative); and b) to demonstrate that the ABA paradigm should not be used to study cognitive deficits in anorexia nervosa. The authors should also discuss whether cognitive deficits were preferentially seen in rats that exhibited severe weight loss in the ABA paradigm. The failure to observe a correlation would undermine the argument that this model can be used to study AN.

2) Please follow the suggestions of reviewers with regards to using the correct terminology, and making it clearer what exactly was done at what point by including more details in the methods as well as which parameters were most relevant to each analysis.

3) Please ensure your manuscript complies with the *eLife* policies for statistical reporting: https://reviewer.elifesciences.org/author-guide/full "Report summary statistics (e.g., t, F values) and degrees of freedom, exact p-values, and 95% confidence intervals wherever possible. These should be reported for all key questions and not only when the p-value is less than 0.05.

*Reviewer #1 (Recommendations for the authors):*

With regards to methodological details and the general presentation of results etc in the text, I have a few suggestions:

1. Please reduce the number of acronyms, it really makes it more difficult to understand when scrolling back constantly to try and remember what things mean. For instance, I don't think "anorexia nervosa" needs to be abbreviated, nor does "pairwise discrimination" and several others.

2. Some of the terminology used is incorrect or needs further definition. Specifically:

– Page 6 "Conditioned reinforcing stimuli." Are these actually conditioned reinforcers? Do the animals respond to the stimuli and not to actual outcomes at some point? If not, I suggested simply calling them "conditioned stimuli" instead.

– Also Page 6: "Negative reinforcers" – Page 6. This terminology refers to the removal of aversive stimuli. Is that what is happening here? Are they seeing the time-out period as a removal of the low tone? Is the low tone aversive? If not, then I suggest rewording this.

3. I suggest including more detail in the methods. For instance, on Page 7 – what did the stages, beginning with habituation, actually, consist of? What stimuli were presented, and what happened when the rats touched the screen? Likewise for all other experimental sections. What reward was used? What were the consequences of animals' actions? It would be good to include enough detail so that the reader doesn't have to refer to published studies to understand what happened.

4. Page 10 – I suggest putting some of these data into the main figures for two reasons (a) it's kind of weird kicking off the Results section by describing results that are entirely in the supplemental figures, and (b) these do offer important controls for the rest of the experimental results – for instance, the effects of ABA exposure on visual operant learning. I suggest the authors pick a couple of key graphs from these figures and add them to Figure 1.

5. I suggest using the less stigmatising term "substance use disorders" as opposed to "addictive disorders".

6. In the abstract "that" should be "than", Line 283 – "be" should be "by", and Line 429 – onto should be "on to" I believe.

– I suggest the authors include some justification as to why only female animals were used for the study on Page 5. Is it because female rats more readily develop ABA? Or for translational purposes? Both?

– I'm not sure what the purpose of synchronising the oestrous cycle of the rats is. In fact, I can see the opposite argument: if the oestrous cycle is random for the rats, then it is unlikely to have had any effect on behavioural outcomes. I suggest including some justification for this also. Page 5

– Please could the authors clarify the experimental numbers? According to Page 6, there are three experiments, n = 20, n = 36, and n = 24. Then on page 8, it stages that 8 animals were excluded and there was a final n = 22. Which experiments were these animals excluded from and how was there a final sample size of n = 22? None of the starting sample sizes are equal to 22 if 8 are excluded from them.

– The discussion point (Lines 454-456) about goal-directed control is an interesting one. I note that compulsion has been argued to be a form of extreme goal-directed control by Hogarth in Neuropsychopharamocology in 2020 (https://www.nature.com/articles/s41386-020-0600-8). We (Bradfield et al., 2017) have also argued that compulsion is the repeated, goal-directed selection of a habit (Bradfleld, L., Morris, R., & Balleine, B. W. (2017). OCD as a failure to integrate goal-directed and habitual action control. In C. Pittenger (Ed.), Obsessive-compulsive disorder: Phenomenology, pathophysiology, and treatment (pp. 343-352). Oxford University Press). The authors may want to take this into consideration.

*Reviewer #2 (Recommendations for the authors):*

I have only a few concerns.

The main concern is the lack of a clear explanation/hypothesis of why prior training in reversal learning makes rats resistant but the first experiment appeared to show the opposite result, increased susceptibility in the rats with better reversal learning performance. Some explanation for this apparent difference is needed.

Another issue is whether all the rats in the first experiment entered the ABA at the same age. Older animals, even by a few days, show more resistance to the ABA paradigm. Did the poorer performance in the touch screen tasks result in these animals being older at the time of ABA? Could this explain the difference in susceptibility?

Please provide a reference to the statement on line 77 that manual transfer to a test chamber affects social behavior.

In the experiment examining the effects of prior ABA exposure, were both susceptible and resistant rats tested for cognitive flexibility? Were there differences?

*Reviewer #3 (Recommendations for the authors):*

1. Figure 1 should be moved to Supplementary Figures. The main figures already include diagrams that provide a good overview of the relevant experimental design.

2. Supplementary Figure 3 should be removed. The distinct patterns observed are not discussed in the text, nor are they utilized in subsequent analyses. Mentioning that half had unlimited access and the other half only had access in the dark phase is extraneous, since these groups are not parsed in the graphs.

3. The data shown in Supplementary Figure 4 are central to the manuscript and should be presented as a main Figure. However, the claim that "…R1 was more difficult to learn than the initial pairwise discrimination" (line 281) is not supported by the statistics shown.

4. Figures 2 and 4 present analyses of many different parameters; a subset shows significant differences, but these are generally not shared across the two figures. While it is important to perform analyses in multiple ways when developing a new behavioral testing system, its validation as a research tool should include clear guidelines about which parameters and analyses are most informative. This will permit pre-registration of studies using this system in the future, which would increase the transparency and rigor of the research and avoid the appearance of p-hacking.

5. In Figure 4, comparisons between those that did and did not learn (L-N) should be moved to Supplementary Figures. While the numbers of non-learning ABA naive mice are too small for meaningful statistical analyses, the differences seem to be driven by learning status more than ABA exposure. It is surprising that the authors did not assess whether susceptibility vs. resistance to ABA impacted learning in Figure 4, since this was a major theme of the paper.

6. The authors propose that learning deficits in ABA-exposed rats are a useful model of AN. The assessment of cognitive behaviors in rats with restricted access to food is critical to validating the utility of this model.

---

## [Author Response]

Essential revisions:1) We request the addition of a control group with restricted access to food in order to validate that this model can be used to study anorexia. Even if the data argue against the validity of the ABA model to study cognitive inflexibility in anorexia nervosa, the manuscript would still be valuable a) As a methods paper (that includes a detailed discussion and guidance regarding parameters that are most informative); and b) to demonstrate that the ABA paradigm should not be used to study cognitive deficits in anorexia nervosa. The authors should also discuss whether cognitive deficits were preferentially seen in rats that exhibited severe weight loss in the ABA paradigm. The failure to observe a correlation would undermine the argument that this model can be used to study AN.

This missing control group was an oversight on our part and has now been included to show that restricted food access alone does not cause the level of impairment in reversal learning seen following exposure to ABA conditions. These animals received exactly the same food access period as ABA rats, across the same number of days, and exhibited what could be described as “mild” weight loss (see updated Figure 4 and figure supplements). With the considerations about operant task type and the potential role of negative reinforcement driving maladaptive behaviours expressed by ABA rats highlighted in the discussion, we believe that ABA remains a valuable tool to study the biological underpinnings of cognitive deficits seen in anorexia nervosa. With respect to the *severity* of weight loss in ABA rats being correlated with preferential cognitive deficits, the narrow weight loss range in ABA (i.e. 20-24%), necessary for adhering to ethical guidelines, precludes an informative evaluation of this. However, the observation that mild weight loss (average 9.01%) in food-restricted control rats did not impair reversal learning to the same extent suggests that the more severe weight loss exhibited by ABA rats was a key driver of cognitive impairment. This supports the argument that the ABA model can be used to study cognitive deficits in AN.

2) Please follow the suggestions of reviewers with regards to using the correct terminology, and making it clearer what exactly was done at what point by including more details in the methods as well as which parameters were most relevant to each analysis.

Terminology has been updated as indicated. Additional methodological and analytical details have been included in the revised manuscript for clarity (see details in responses to individual reviewers below).

3) Please ensure your manuscript complies with the eLife policies for statistical reporting: https://reviewer.elifesciences.org/author-guide/full "Report summary statistics (e.g., t, F values) and degrees of freedom, exact p-values, and 95% confidence intervals wherever possible. These should be reported for all key questions and not only when the p-value is less than 0.05.

We wholeheartedly support full transparency of statistics, not only for when the p value is less than 0.05. To facilitate easy understanding of the statistical evaluations made, instead of including one table with statistical comparisons from all figures as a Supplementary File (as we did in our initial submission), we have now included tables of statistics as source data files to accompany each figure and figure supplement package. We have also updated the text to include all p values for completeness.

Reviewer #1 (Recommendations for the authors):With regards to methodological details and the general presentation of results etc in the text, I have a few suggestions:1. Please reduce the number of acronyms, it really makes it more difficult to understand when scrolling back constantly to try and remember what things mean. For instance, I don't think "anorexia nervosa" needs to be abbreviated, nor does "pairwise discrimination" and several others.

We have removed all acronyms from the main text, including anorexia nervosa (AN), pairwise discrimination (PD), reversal 1 (R1), food anticipatory activity (FAA) and running wheel activity (RWA). The only acronym we are keeping in the text is ABA.

We have, however, decided to keep the acronyms in the figure panels, as non-abbreviated forms of these make the figures less readable and considering they are clearly defined in figure legends we believe this is appropriate.

2. Some of the terminology used is incorrect or needs further definition. Specifically:– Page 6 "Conditioned reinforcing stimuli." Are these actually conditioned reinforcers? Do the animals respond to the stimuli and not to actual outcomes at some point? If not, I suggested simply calling them "conditioned stimuli" instead.– Also Page 6: "Negative reinforcers" – Page 6. This terminology refers to the removal of aversive stimuli. Is that what is happening here? Are they seeing the time-out period as a removal of the low tone? Is the low tone aversive? If not, then I suggest rewording this.

Completely agree with both points and have updated the text accordingly.

3. I suggest including more detail in the methods. For instance, on Page 7 – what did the stages, beginning with habituation, actually, consist of? What stimuli were presented, and what happened when the rats touched the screen? Likewise for all other experimental sections. What reward was used? What were the consequences of animals' actions? It would be good to include enough detail so that the reader doesn't have to refer to published studies to understand what happened.

Because of the sheer amount of detail involved in these methods including the different training and testing session stimuli and stages, we opted in the original submission to include all of this information in the form of a graphical timeline, a supplementary table and a flowchart figure. We have now expanded on these resources and positioned them as additional resources to the main figures to enable easier reading and reference to these details. The graphical schematic images for pretraining are in the graphical timeline figure, the table has been updated to say ‘illuminated square’ rather than ‘image’ for pretraining stimuli, and to clarify no effect for touching blank screen for stages 2-4, and the flowchart has had slight modifications (ITI numbers added in in place of X placeholder).

4. Page 10 – I suggest putting some of these data into the main figures for two reasons (a) it's kind of weird kicking off the Results section by describing results that are entirely in the supplemental figures, and (b) these do offer important controls for the rest of the experimental results – for instance, the effects of ABA exposure on visual operant learning. I suggest the authors pick a couple of key graphs from these figures and add them to Figure 1.

We struggled with the most appropriate order for main and supplementary figures for this very reason – although the main “thrust” of the paper is exploring the bidirectional effects of ABA exposure and reversal learning, we felt it still necessary and informative to include the system validation and optimisation data (as duly noted by this reviewer). We have now reshuffled the results so that the first data figure (Figure 2) deals with these details, so that these can be used as baseline data for the rest of the experimental results. We have also reshuffled all supplementary files into figure supplements to support main figures based on guidance from the *eLife* editorial team.

5. I suggest using the less stigmatising term "substance use disorders" as opposed to "addictive disorders".6. In the abstract "that" should be "than", Line 283 – "be" should be "by", and Line 429 – onto should be "on to" I believe.

These have been fixed, appreciate the thorough review.

– I suggest the authors include some justification as to why only female animals were used for the study on Page 5. Is it because female rats more readily develop ABA? Or for translational purposes? Both?

Indeed, both reasons indicated by the reviewer justify the use of female animals exclusively in these studies. This rationale has now been explicitly stated on Page 8 *with the following text: “Young female rats were used in these studies because they are particularly vulnerable to developing the ABA phenotype, a feature that is incompletely understood but has translational relevance to the increased prevalence of anorexia nervosa in young women”.*

– I'm not sure what the purpose of synchronising the oestrous cycle of the rats is. In fact, I can see the opposite argument: if the oestrous cycle is random for the rats, then it is unlikely to have had any effect on behavioural outcomes. I suggest including some justification for this also. Page 5

It has long been known that both wheel running and food intake fluctuate over the oestrous cycle in female rats (e.g. https://pubmed.ncbi.nlm.nih.gov/2602468/) and cycling is also shown to be disrupted by ABA conditions (https://pubmed.ncbi.nlm.nih.gov/14637226/). Therefore, in order to ensure that susceptibility to ABA is not confounded by cyclic fluctuations in either running activity or feeding, we aim to ensure that animals are largely synchronised at the commencement of ABA conditions. This rationale has now been explicitly defined on Page 8.

– Please could the authors clarify the experimental numbers? According to Page 6, there are three experiments, n = 20, n = 36, and n = 24. Then on page 8, it stages that 8 animals were excluded and there was a final n = 22. Which experiments were these animals excluded from and how was there a final sample size of n = 22? None of the starting sample sizes are equal to 22 if 8 are excluded from them.

This was a simple error on our part – the initial total group size for the experiment in which we first tested flexible learning and subsequently exposed animals to ABA conditions was 30 (not 36), meaning that after exclusions (as described) the final sample size was now n=22.

– The discussion point (Lines 454-456) about goal-directed control is an interesting one. I note that compulsion has been argued to be a form of extreme goal-directed control by Hogarth in Neuropsychopharamocology in 2020 (https://www.nature.com/articles/s41386-020-0600-8). We (Bradfield et al., 2017) have also argued that compulsion is the repeated, goal-directed selection of a habit (Bradfleld, L., Morris, R., & Balleine, B. W. (2017). OCD as a failure to integrate goal-directed and habitual action control. In C. Pittenger (Ed.), Obsessive-compulsive disorder: Phenomenology, pathophysiology, and treatment (pp. 343-352). Oxford University Press). The authors may want to take this into consideration.

We would like to thank the reviewer for the suggestion of including a clear definition of a compulsion which we have used in the discussion to facilitate an elaboration on the subgroups of individuals suffering from anorexia nervosa that appear to be more habit-driven versus goal-directed in reward-related behaviours as well as the note from the comment above about outcome-specific reversal learning in ABA rats.

Reviewer #2 (Recommendations for the authors):I have only a few concerns.The main concern is the lack of a clear explanation/hypothesis of why prior training in reversal learning makes rats resistant but the first experiment appeared to show the opposite result, increased susceptibility in the rats with better reversal learning performance. Some explanation for this apparent difference is needed.

We appreciate this concern and acknowledge the disconnect was not explored sufficiently in the previous submission. There are 2 parts to this explanation that have now been included in the discussion text of the manuscript in the appropriate paragraphs.

The first part relates to why those rats that were fastest to learn the reversal task went onto be susceptible to weight loss in ABA. What we believe to be the case, although this has not yet been empirically shown, is that food restriction evoked hyperactivity in ABA, one of the critical drivers of susceptibility to weight loss, is reinforced through the alleviation of negative affect (stress, hunger etc). Performance on the reversal task is also driven by negative feedback (i.e. previously learned actions are not rewarded until the animal adapts its behaviour to select the previously unrewarded image). Rats that went onto be susceptible to ABA required less negative feedback to learn the reversal task, therefore it seems plausible that they more quickly learned the reinforcing properties of wheel running during restricted food access, engage in more running behaviour and ultimately become susceptible to ABA.

Regarding why exposure to cognitive testing produced more resistant rats, seemingly at odds with the above result, we believe this relates to the procedural aspects of training rather than to task performance *per se*. Regardless of task performance, rats that underwent cognitive testing received sucrose rewards over multiple test days and were subject to mild food restriction as per standard operant test protocols. Both of these procedural aspects may have caused the shift toward resistance compared to animals that were experimentally naïve prior to ABA exposure. It should also be noted that the test group that were exposed to ABA first and then tested on the reversal task were surprisingly mostly susceptible (21/22 rats) a much higher proportion than we usually see on a cohort-to-cohort basis.

Another issue is whether all the rats in the first experiment entered the ABA at the same age. Older animals, even by a few days, show more resistance to the ABA paradigm. Did the poorer performance in the touch screen tasks result in these animals being older at the time of ABA? Could this explain the difference in susceptibility?

We understand this reviewers concern regarding the age at which ABA commenced impacting on susceptibility. It is true that animals entered the ABA paradigm once they had learned the reversal task, such that the faster learners would commence ABA exposure some days earlier than slower learners. However, there was no significant difference in the age at which ABA commenced between Susceptible and Resistant subpopulations (see additional resource for Figure 3 —figure supplement 2), indicating that age was not a contributing factor to the difference in susceptibility. Text in methods has been edited for clarity.

Please provide a reference to the statement on line 77 that manual transfer to a test chamber affects social behavior.

Our intention with this statement was more a comment on a typical housing situation in which animals are pair-housed during operant testing and when transferred manually to conventional test chambers, the home-cage social dynamics are disrupted. We appreciate that this is somewhat outside of the scope of the statement and has not been empirically tested, so we have removed this statement for clarity.

In the experiment examining the effects of prior ABA exposure, were both susceptible and resistant rats tested for cognitive flexibility? Were there differences?

As noted in Figure 5B, there was only 1 animal resistant to ABA in the prior exposure group, therefore data from all animals was pooled. In terms of where this animal sits in reversal learning rate, we have prepared bar graphs of performance measures with the resistant animal highlighted in red to show that while this resistant rat was among the fastest of the ABA exposed animals to acquire the task, it was not a statistical outlier (Figure 4—figure supplement 2). We have been careful throughout the manuscript to describe effects of “exposure to ABA conditions” rather than “susceptibility to ABA” on learning outcomes based on these data.

Reviewer #3 (Recommendations for the authors):1. Figure 1 should be moved to Supplementary Figures. The main figures already include diagrams that provide a good overview of the relevant experimental design.

Because of the format of figures required by the *eLife* editorial team, we were unable to include this as a Supplementary File these are reserved for long lists of strains and plasmids, lengthy descriptions of algorithms and mathematical proofs. Moreover, based on the overall comment by this reviewer about the strength of this manuscript as a “methods paper” and in agreement with the editorial team, we have included the methods of this manuscript prior to the results, and to complement this all figures pertaining to the methodologies employed are presented in Figure 1 and associated figure supplements, as this made the most sense for the flow of the article.

2. Supplementary Figure 3 should be removed. The distinct patterns observed are not discussed in the text, nor are they utilized in subsequent analyses. Mentioning that half hadunlimited access and the other half only had access in the dark phase is extraneous, since these groups are not parsed in the graphs.

Agree and we have removed this supplementary information.

3. The data shown in Supplementary Figure 4 are central to the manuscript and should be presented as a main Figure. However, the claim that "…R1 was more difficult to learn than the initial pairwise discrimination" (line 281) is not supported by the statistics shown.

The data in Supplementary Figure 4 are now presented as the first data figure (Figure 2) and the issue with this claim has been noted, revised and amended in the text relating to this result to now read:

“We also confirmed that subsequent reversals were progressively easier to learn than the initial reversal with fewer session required to reach performance criterion (first reversal > second reversal: p=.0099; first reversal > third reversal: p=.0070; Figure 2B).”

4. Figures 2 and 4 present analyses of many different parameters; a subset shows significant differences, but these are generally not shared across the two figures. While it is important to perform analyses in multiple ways when developing a new behavioral testing system, its validation as a research tool should include clear guidelines about which parameters and analyses are most informative. This will permit pre-registration of studies using this system in the future, which would increase the transparency and rigor of the research and avoid the appearance of p-hacking.

We understand the concern and note that the number of trials (i.e. action-outcome associations) to performance criterion is the most informative parameter to indicate learning rate, in addition to the trial response types (correct, incorrect, omitted), which shows the amount of reinforcement required for learning. However, a minimum number of both trials and sessions are required to be met for progression criteria, a factor that necessitates the presentation of both parameters. We also believe that the number of sessions animals required to reach criterion is also important for describing the learning rate and complements the trials measures, because it incorporates consolidation as well as allows for comparison with traditional methods of touchscreen operant testing. We agree that the number of days to criterion is largely superfluous for understanding cognitive performance, particularly because the number of sessions allowed per day is determined by the experimenter. Thus, while we have retained the “days to criterion” measure in the validation of the novel system and method figure (Figure 2), we have removed these data from other figures. We have also removed the ratio of correct/non-correct graph from Figure 3 as it in essence recapitulates the correct and non-correct trials in their raw data form.

5. In Figure 4, comparisons between those that did and did not learn (L-N) should be moved to Supplementary Figures. While the numbers of non-learning ABA naive mice are too small for meaningful statistical analyses, the differences seem to be driven by learning status more than ABA exposure. It is surprising that the authors did not assess whether susceptibility vs. resistance to ABA impacted learning in Figure 4, since this was a major theme of the paper.

We have moved the comparisons between those that did and did not learn from this figure to a figure supplement (Figure 4—figure supplement 1) as we agree with the Reviewer’s conclusion that the differences here are likely driven by learning status. We did not assess whether susceptibility versus resistance to ABA impacted these outcomes because only one animal in this cohort was resistant to ABA prior to cognitive testing (a curious feature of this particular cohort, but not beyond the scope of the reported literature regarding cohort-to-cohort variability in susceptibility profiles of Sprague-Dawley rats). We have prepared a subset of graphs from this figure panel in which the resistant animal is highlighted in red (Figure 4—figure supplement 2) to demonstrate this animal did not perform differently to susceptible animals, but rather the exposure to ABA conditions independent of amount of weight loss, was the driver of poor performance. This view of understanding the effects of ABA experience, rather than weight loss per se is again not uncommon in the field (see https://pubmed.ncbi.nlm.nih.gov/33368559/ for example) and is supported by the new experimental details that show a matched period of food restriction does not induce severe weight loss or the same extent of impairment in cognitive performance.

6. The authors propose that learning deficits in ABA-exposed rats are a useful model of AN. The assessment of cognitive behaviors in rats with restricted access to food is critical to validating the utility of this model.

100% agree and feel somewhat sheepish to have overlooked this important control experiment. This has now been completed and incorporated into Figure 4 and associated resources.